# Exploring Spatially Non-Stationary and Scale-Dependent Responses of Ecosystem Services to Urbanization in Wuhan, China

**DOI:** 10.3390/ijerph17092989

**Published:** 2020-04-25

**Authors:** Yan Zhang, Yanfang Liu, Jiawei Pan, Yang Zhang, Dianfeng Liu, Huiting Chen, Junqing Wei, Ziyi Zhang, Yaolin Liu

**Affiliations:** 1School of Resource and Environmental Sciences, Wuhan University, 129 Luoyu Road, Wuhan 430079, Chinayaolin610@yeah.net (Y.L.); 2Collaborative Innovation Center for Geospatial Information Technology, Wuhan University, 129 Luoyu Road, Wuhan 430079, China; 3Key Laboratory of Geographic Information System, Ministry of Education, Wuhan University, 129 Luoyu Road, Wuhan 430079, China; 4College of Urban Economics and Public Administration, Capital University of Economics and Business, Beijing 100070, China

**Keywords:** Ecosystem services, Urbanization, Non-stationary relationships, Scale dependent

## Abstract

Ecosystem services (ESs) are facing challenges from urbanization processes globally. Exploring how ESs respond to urbanization provides valuable information for ecological protection and urban landscape planning. Previous studies mainly focused on the global and single-scaled responses of ESs but ignored the spatially heterogenous and scale-dependent characteristics of these responses. This study chose Wuhan City in China as the study area to explore the spatially varying and scale-dependent responses of ESs, i.e., grain productivity, carbon sequestration, biodiversity potential and erosion prevention, to urbanization using geographically weighted regression (GWR). The results showed that the responses of ESs were spatially nonstationary evidenced by a set of local parameter estimates in GWR models, and scale-dependent indicated by two kinds of scale effects: effect of different bandwidths and effect of grid scales. The stationary index of GWR declined rapidly as the bandwidth increased until reaching to a distance threshold. Moreover, GWR outperformed ordinary least square at both grid scales (i.e., 5 km and 10 km scales) and behaved better at finer scale. The spatially non-stationary and scale-dependent responses of ESs to urbanization are expected to provide beneficial guidance for ecologically friendly urban planning.

## 1. Introduction

Ecosystem services (ESs) refers to “goods and services humans obtain from ecosystems directly or indirectly”. These ESs range from those support human survival (e.g., clean water) to those enhance health and well-being (e.g., climate regulation and outdoor recreation). Millennium ecosystem assessment (MA) [1] categorized ESs into four types: provision services, regulating services, supporting services and cultural services. Since MA, an increasing number of efforts have been undertaken to develop our knowledge on this concept. Fisher [2] claimed that we should distinguish “ends” and “means” of ESs if we want to operate it in policy making. Boyd and Banzhaf [3] introduced a term “final ecosystem services” and defined it as “components of nature directly enjoyed, consumed or used to yield human well-being”. The Common International Classification of Ecosystem Services (CICES) [4] then developed a hierarchy to classify final ESs, which is composed by levels of “section, division, group, class and class type”. Moving from top to down, the ‘services’ become more specific but remain nested within the broader categories above them. Recently, a program Intergovernmental Science-Policy Platform on Biodiversity and Ecosystem Services (IPBES) proposed a novel conceptual model “nature’s contributions to people” (NCP) [5] to overcome the drawback of MA-based ESs concept that failed to engage perspectives from social sciences. This framework summarized three categories of NCPs including material NCPs, non-material NCPs and regulating NCPs. Debates on ESs deepen our understanding on this concept and benefit its operations in landscape planning, management and decision-making.

Ecosystem services are extensively affected by the socio-ecological environment [6]. Knowledge on the driving factors of ESs is crucial to the design of environmental policy. Driving factors can be mainly divided into two types: ecological environment factors [7,8] and social factors [9]. Ecological environment factors mainly refer to the natural factors, like climate, topography and soil, which are the basis to form ESs. Social factors refer to social and economic activities, like population, economy, policy, science and technology and culture that affect the change of ESs. In the age of escalating industrialization and upgrading urbanization, human factors pose increasing impacts on ESs provisioning. Human activities modify land habitat, alter ecosystem structure and change the biogeochemical cycle, and thus greatly change the formation and supply of ecological services.

Among all social drivers, urbanization is recognized as a main determinant of ESs decline [10]. For instance, urban land transformation modifies the quantity and quality of forests and thus reduces timber supply [11]. The runoff in urban surface increases for the reason of the decrease of interception, evapotranspiration and infiltration, and thus leads to the water quality degradation and water yield increment [12], Baró, Gómez-Baggethun [13] quantified how six ESs (i.e., crops, livestock, climate regulation, air purification, erosion control and outdoor recreation) varied along with urban-rural gradient and revealed that all ESs significantly improved as the distance to the city center increased. Due to the vital roles that ESs play in sustaining human organisms, the degradation of them would greatly threaten human well-being and health [14]. For example, the large-area loss of cultivated land during urban expansion would threaten the food security of a society. In addition, the removal of natural vegetation would reduce the ecosystem’s capacity to capture air pollution and then harm public health [15].

ESs are facing increasing pressure from the rapid urbanization process in China since its reform and opening up policy in 1978. According to relevant report, the urbanization rate increased from 17.9% to 56.1% from 1978 to 2015 in China. Meanwhile, urban land expansion is faster than population growth [16]. Research on relationships between ESs and urbanization has been conducted at various space and time scales in China [17,18,19,20,21,22]. Qiu, Li [23] explored the vulnerability of ESs provisioning to urbanization in 31 provinces of China during 1980–2010 and revealed that vulnerability was higher in the eastern provinces; in addition, vulnerability increased in the eastern and central provinces. Peng, Tian [18] utilized linear regression to investigate the relationship between total ESs and urbanization in Beijing, and found a globally negative correlations between them. Wan, Ye [24] identified an irregular inverse “U” correlation between ESs and urbanization during 1990–2011 in a mineral resource-based city. To overcome the drawback of previous studies that ignored the spatial issue in ESs-urbanization interaction, Zhang, Liu [22] adopted spatial techniques (i.e., spatial lag/error model) to model the spatial relationships between six ESs and urbanization, and revealed that urbanization damaged ESs generally. However, these researches mainly focused on global relationships but ignored the ESs-urbanization interaction at local space, which may tell more for urban landscape planning and management. In this context, geographically weighted regression (GWR) model, which is capable to address spatially non-stationary relationships, should be further adopted [25]. In addition to spatial non-stationarity, present studies lacked discussions on the scale dependency in the spatially non-stationary relationships between ESs and urbanization. In fact, distributions and dynamics of ESs are usually scale dependent. Multiple scale analysis would help to better understand the causal relationships between ESs and urbanization across scales, and thus improve the accuracy of modeling results.

This study, with a case study of Wuhan in China, aims to explore the non-stationary and scale-dependent relationships between ESs and urbanization. To be specific, this research attempts to: (1) map the changes of ESs and identify clustering feature of these changes; (2) explore spatially non-stationary responses of ESs to urbanization by GWR; (3) conduct multi-scale analysis on the spatially varying responses to characterize scale effects. We expected to provide policy implications for urban planning and ecological conservation.

## 2. Materials and Methods

### 2.1. Study Area

Wuhan city, located at 29°58′–31°22′ N and 113°41′–115°05′ E, is the capital city of Hubei province and is the biggest city in central China (Figure 1). It covers 8569.15 km^2^ and subordinates 13 districts. The altitude is between 19.2 and 873.7 m, and the climate is humid north subtropical monsoon. Wuhan is abundant of natural habitats: wetlands and forests covers 39.54% and 11.4% of total territory.

As the biggest city in central China, Wuhan has experienced rapid socio-economic development since the 2000s. From 1990 to 2015, the population in Wuhan increased from 3.8 million to 6.0 million, while the urban land expanded from 260.91 km^2^ to 503.71 km^2^. Rapid urban expansion in Wuhan has damaged its natural ecosystems. According to Wang [26], 803.64 km^2^ of cropland has been occupied by urban expansion during the period of 1990–2015. Meanwhile, many natural habitats have been replaced by artificial and impervious surfaces. For instance, 28 km^2^ of wetland have been developed during 1990–2013, including 77% of Shahu Lake and 52% of Nanhu Lake [27]. Facing this situation, government in Wuhan is putting much effort into coordinating its development with the ecological environment. In this context, it is meaningful to explore how ESs respond to urbanization and specify their implications for ecologically friendly urban planning in Wuhan.

### 2.2. Data Source and Preprocessing

Multi-source data were integrated and utilized in our study, including (1) LULC data in 2005 and 2015 in 30 m × 30 m raster format obtained from the Wuhan Natural Resources Bureau. LULC raster dataset was classified into six land use types, i.e., cultivated land, forest, grassland, waterbodies, urban land and other land. (2) Digital evaluation model data (DEM) in 30 m × 30 m raster format provided by the Wuhan Planning and Research Institute. (3) Meteorological datasets (including air temperature, rainfall and radiation) in 2005 and 2015 obtained from China Meteorological Data Sharing Service System (http://data.cma.cn/site/index.html). The original datasets were in point shapefile format and then interpolated to the entire area using ordinary kriging technique. (4) Normalized difference vegetation index (NDVI) data extracted from MODIS/Terra 250 m 16-day product MOD13Q1. (5) Soil datasets were extracted from China Soil Map Based Harmonized World Soil Database (v1.2) provided by Scientific Data Center of Cold and Arid Regions (http://westdc.westgis.ac.cn/). (6) Statistical grain yield data in 2005 and 2015 were extracted from the Wuhan Statistical Yearbook, provided by Wuhan Municipal Bureau of Statistics. (7) Maps of population density in 2005 and 2015 were obtained from the geographic information monitoring platform (http://www.dsac.cn/). (8) The road network in Wuhan was obtained from the Wuhan Planning and Research Institute.

### 2.3. Methods

#### 2.3.1. Flowchart of the Method

The methods include five main components (Figure 2): assessment of ESs, measurement of urbanization, variables preparation for regression, spatially non-stationary responses of ESs to urbanization and scale effects in the spatially non-stationary relationships. The main steps of the method were introduced as follows:

Step 1: We assessed ESs at both 2005 and 2015.

Step 2: We evaluated urbanization at both 2005 and 2015.

Step 3: We extracted ESs changes and urbanization during 2005–2015 and then prepared independent and dependent variables for regression at 5 km and 10 km grid scales.

Step 4: We conducted OLS and GWR to explore the responses of ESs to urbanization at both scales, with OLS utilized for the global responses while GWR for spatially non-stationary responses. GWR results, including sensitivity of non-stationarity to bandwidth, spatially varying responses, and model performance and its comparison with OLS’s, were elaborated in detail.

Step 5: We discussed two kinds of scale effects in the spatially non-stationary relationships between ESs and urbanization.

#### 2.3.2. Evaluation of Ecosystem Services and Mapping of Their Changes

Indictors for ESs were selected based on the principles that they were: (1) good representatives of the ecological benefits humans require in the study area; (2) vulnerable to urbanization and; (3) available to be quantified based on the existing models and data. The following section details the definition and evaluation of the selected ESs. Table A1 in the Appendix A provides the detailed calculation process for the six ESs.

Grain product (GP) is a service manifesting the ability of a land area to provide grains (e.g., rice, wheat, millet and soybean, etc.), which is especially important to food security in a region. It was assessed by the empirical regression between grain product and the vegetation condition index (VCI) [28]. In this method, the statistical GP value in a region was downscaled to each cultivated land grid according to the ratio of the grid’ VCI to the total VCI value.

Carbon storage (CS) represents the amount of organic material carbon sequestrated by green vegetations during a certain time period. It plays a key role in biologic carbon cycle and sustainable development of terrestrial ecosystem. Net primary productivity (NPP) was assessed as a proxy variable for CS using the Carnegie–Ames–Stanford Approach (CASA) model [29]. This method multiplies the absorbed photosynthesis active radiation (APAR) and a light use efficiency parameter (ε) to calculate NPP.

Biodiversity is usually not regarded as a service but rather is treated as the basis for ecosystem services [1]. Some studies have evaluated biodiversity by using species indicators [30,31]. Different from them, we used an indicator habitat quality as the proxy variable of biodiversity potential (BP) because lacking species data. Habitat quality was estimated using the InVEST tool [32]. In this tool, habitat quality of a patch was calculated according to its suitability to natural habitat as well as its exposure to anthropocentric interfere [33,34].

Erosion prevention (EP) is a service to represent the ability of an ecosystem to maintain soil in resistance to transportation and deposition of exogenic forces such as wind, water and freeze-thaw damages, etc. We utilized the opposite number of soil erosion intensity as the proxy for erosion prevention ability of an ecosystem [35]. The soil erosion intensity was estimated by universal soil loss equation (USLE).

After each individual ES was obtained, changes of them during 2005–2015 were obtained by cross-comparing the digital maps of ESs in 2015 and 2005. To examine the scale effect, we displayed the spatial patterns of ESs changes at two grid scales: 5 km and 10 km grids. Analysis on these two scales enabled us to detect spatially heterogenous patterns of ESs changes at small land units but avoid the computing burden and cartographic difficulty at finer scales (e.g., 1 km patches).

Spatial autocorrelations of ESs changes were examined by Moran’s I method, which reveals whether and to what extent spatial aggregations exist across the entire area [36]. The values of Moran’s I are ranged between −1 and 1, with positive values indicating a spatial clustering pattern, while negative values for a spatial dispersion pattern. The statistical significance of Moran’s I is usually examined by a *p* value: *p* < 0.1 indicates a significant spatial autocorrelation. We conducted Moran’s I analysis in GeoDa 1.12 (http://geodacenter.github.io/) software.

#### 2.3.3. Measurement of Urbanization

Urbanization can be manifested at different aspects, for example, population growth, economic advance, life-style modification and urban land expansion [37]. Three indicators were selected to characterize urbanization referring to previous works [18,38], including population growth (PG), urban land expansion (ULE) and distance to major roads (Dis_road). PG and ULE are two urbanization intensity indicators and Dis_road represents urban influence gradient. Changes of ESs in response to these urbanization indicators can reflect the degree of anthropogenic influences on ecosystems.

Indicators for urbanization during 2005–2015 were calculated at both grid scales. The two urbanization intensity indicators for each grid were calculated using Equations (1) and (2). The distance buffers of major roads were generated using the DISTANCE method in ArcGIS 10.5.
(1)$$PG_{i}=PD_{i,t+n}-PD_{i,t}$$
(2)$$ULE_{i}=\frac{ULA_{i,t+n}-ULA_{i,t}}{TA_{i}}\times 100\%$$
where $PG_{i}$ and $ULE_{i}$ represent PG and ULE in grid *i*, respectively. $PD_{i,t}$ and $PD_{i,t+n}$ represent population density in grid *i* at year *t* (i.e., year 2005) and *t* + *n* (i.e., year 2015), respectively. $ULA_{i,t}\;\mathrm{and}\;ULA_{i,t+n}$ are urban land area in grid *i* at year *t* and *t* + *n*, respectively. $TA_{i}$ signifies the total area of grid *i*.

#### 2.3.4. Geographically Weighted Regression

We implemented ordinary least squares regression (OLS) and geographically weighted regression (GWR) to explore how ES respond to urbanization. Developed from OLS, the spatial technique GWR has ability to model spatially non-stationary relationships [39]. It generates a set of location-based estimates, including local R^2^, local parameters and local model residuals, and displays the spatial patterns of local estimates [40]. Mathematical equations for OLS and GWR models were presented in Equations (3) and (4):(3)$$y_{i}=\beta_{0}+\overset{k}{\underset{i=1}{\sum }}\beta_{i}x_{i}+\varepsilon$$
where y is the value of a certain ecosystem service; $x_{i}$ denotes the i-th urbanization indicator; k is the number of urbanization indicators (*k* = 1 in this case); $\beta_{0}$ is the intercept coefficient and $\beta_{i}$ is the coefficient for the i-th urbanization indicator; ε denotes stochastic error.
(4)$$y_{i}=\beta_{0}\left(u_{j},v_{j}\right)+\overset{k}{\underset{i=1}{\sum }}\beta_{i}\left(u_{j},v_{j}\right)x_{ij}+\varepsilon_{j}$$
where $u_{j}\;\mathrm{and}\;v_{j}$ denote geographic coordinates of location *j*, $\beta_{0}\left(u_{j},v_{j}\right)$ is the intercept parameter at location *j*; $\beta_{i}\left(u_{j},v_{j}\right)$ is the local estimated coefficient for the *i-th* urbanization indicator at location *j*.

A distance decay function is required in GWR to reflect the phenomenon that the near distance observation has a higher impact on the estimation of local parameters at the center location. Gaussian distance is usually adopted in GWR to express the spatial interactions as the decay function, which could be expressed in the following equation:(5)$$w_{ij}=\exp\left(-d_{ij}^{2}/h^{2}\right)$$
where $w_{ij}$ is the weight for location *j* in terms of location *i*,$\;d_{ij}$ is the distance from location *j* to *i*, *h* is the kernel bandwidth. When the real distance exceeds the kernel bandwidth, the weight becomes zero. The size of kernel bandwidth will affect the performance of GWR. There are two common types of bandwidth: fixed bandwidth and adaptive bandwidth [40]. The fixed kernel has a constant bandwidth in space [41], while the adaptive kernel can adjust the bandwidth according to the change of data density. In order to obtain the real value of kernel bandwidth and reveal its implications for ecological assessment, we used the fixed kernel. We determined the optimal kernel according to the spatial stationarity index of GWR (see more processing details in Section 2.3.5).

Before implementation of GWR, the value of each individual ES change during 2005–2015 was designated as dependent variable and each individual urbanization indicator during 2005–2015 as independent variable. In other words, GWR was repeatedly operated for each pair of ES change and urbanization indicators. Moreover, GWR was implemented at both grid scales to examine the scale effect in the non-stationary responses of ESs to urbanization.

#### 2.3.5. Measurement of Non-Stationarity and Scale Analysis

A stationarity index was utilized to measure spatial non-stationarity in the interaction between ESs and urbanization referring to the researches of Fotheringham, Charlton [42] and Osborne, Foody [43]. It was calculated by three steps: (1) the interquartile range for standard errors of the GWR coefficients was calculated; (2) twice standard error of the OLS coefficient was obtained; (3) the ratio of these two values was calculated as the stationary index. The stationarity index with value less than one signifies spatial stationarity [44]. The size of kernel bandwidth affects the stationarity index of GWR, so we examined the sensitivity of stationarity index to kernel bandwidth. We then selected the kernel bandwidth that produced relatively stable stationarity index as the optimal for GWR.

The sensitivity of the stationarity index to bandwidth was examined at both grid scales. To be specific, we implemented GWR model repeatedly at 5 km scale with the kernel bandwidth increasing from 5000 to 50,000 m at an interval of 5000 m; we also conducted GWR model repeatedly at 10 km scale with the kernel bandwidth increasing from 10,000 to 100,000 at an interval of 10,000 m. Analysis on the sensitivity of stationarity index to kernel bandwidth was implemented in Python 3.5 environment.

## 3. Results

### 3.1. Spatial Patterns of Ecosystem Services and Biodiversity Changes

ESs changes during 2005–2015 showed uneven distributions across the region and featured “global similarity and local difference” at different scales (Figure 3). For GP changes, negative values were widely distributed in the main grain producing areas, while positive values were scattered in rural regions. For CS changes, negative values occupied the most areas, including the main urban areas, urban peripheries and rural areas. Positive values were sparsely distributed in mountainous areas in the northeast and northwest. For BP changes, negative values were dominant and mainly aggregated in the urban periphery, while positive values of BP changes were also common, mainly in areas away from human settlements in the northeast and southeast. For EP changes, positive values were dominant, mainly concentrated in the plain areas away from urban areas. On the contrary, negative values were clustered in urban fringe.

Table 1 presents the spatial autocorrelations of the changes in four ESs. We can observe that most global Moran’s I values were positive values with statistical significance at the 0.05 level (except that of BP at 5 km grid), implying that clustering distribution of ESs changes was widespread.

### 3.2. Spatial Patterns of Urbanization Indicators

Indicators for urbanization during 2005–2015 showed spatially heterogenous characteristics (Figure 4). For PG, positive values were concentrated in and around urban areas, whereas negative values were sparsely distributed away from urban areas, which indicates that population mitigate from rural to urban areas during this period. For ULE, positive values were concentrated in and around urban areas, whereas negative values were distributed adjacent to core urban areas. Conflict between PG and ULE was detected in rural regions: although population decreased in many rural areas, urban land still increased. This indicates an uncontrolled urban land expansion in outer urban areas of Wuhan during the study period. Spatial distribution of Dis_road showed that low values were aggregated in urban hinterland, whereas high values were distributed in edge of the region.

### 3.3. Responses of Ecosystem Services to Urbanization

#### 3.3.1. Globally Responses of Ecosystem Services by OLS

Coefficients for urbanization indicators generated by OLS were shown in Table 2. Most coefficients were statistically significant at *p* < 0.05, which proves that urbanization was an important driver of ESs change. However, directions and strengths of coefficients varied with urbanization indicators and grid scales. In general, PG and ULE posed negative effects on most ESs, signifying that PG and ULE would cause ESs degradation in general, whereas Dis_road exerted positive effect on all ESs, indicating that regions near to road would have higher risk of ESs degradation. Among four ESs, PG damaged GP and EP most and CS least; ULE damaged GP and EP most and BP least; Dis_road influenced GP and CS most and BP least. Except for EP, urbanization indicators exerted stronger effects on other ESs at finer scale.

#### 3.3.2. Sensitivity of Non-Stationarity to Bandwidth in GWR

Figure 5 presents the sensitivity of stationarity index of GWR in response to bandwidth change. In general, the stationary indexes declined rapidly with increase of bandwidth until reaching to a certain distance threshold. The gradient of decline and stationary thresholds differed with urbanization predicators and grid scales. Compared to the other three urbanization indicators, ULE had not only the sharper descent slope, but also a smaller distance threshold, which indicates that urban expansion influences the ESs changes at smaller spatial scale range. With ULE as predictors, distance thresholds were observed as 15,000 m and 30,000 m at 5 km and 10 km grid scales, respectively. Moreover, compared to finer grid scales, thresholds at coarser grid seemed to be at larger distance. The distance thresholds above which stationarity became relatively flat were chosen as the operational bandwidth in the following GWR regressions.

#### 3.3.3. Spatially Non-Stationary Responses of Ecosystem Services by GWR

Maps of coefficients, local R^2^ and standardized residuals (StdResid) generated from GWR models visualized the spatially non-stationary relationships. Coefficients presented the direction and strength of relationships between ESs and urbanization. Local R^2^, ranging between 0 and 1, tells local model fit: low values indicate poor model performance. Residuals signified the deviations of the predicted values to observed values and standardized residuals are ranged from 0 to 1.

From Figure 6, Figure 7, Figure 8 and Figure 9, we can see, model fits of GWR seemed to be stronger in regions where ESs changes presented heterogenous patterns with their neighboring regions. Stronger model fits were also observed in places where urbanization advanced significantly. In contrast, modeling accuracy of GWR seemed to be lower in areas with homogeneous ESs changes or no obvious urbanization advance. In general, urban surrounding areas obtained bigger local R^2^ values than those in urban centers. Moreover, rural regions in central plain areas obtained bigger local R^2^ values than those in the mountainous areas. The direction and strength of the relationships generated by GWR were also spatially varied. Strongly negative correlations between most ESs changes (i.e., GP, CS and BP) and urbanization aggregated in urban surrounding areas (Figure 6, Figure 7 and Figure 8), which indicate that urbanization bring strongly destructive effects on ESs in these places. On the contrary, weakly negative relationships were detected in rural regions, which indicated that damage of urbanization in these areas was rather weak.

Although OLS detected global relationships between ESs changes and urbanization, some deviated results were identified by GWR at local space. OLS revealed negative correlations of most ESs changes with PG and ULE and positive correlations with Dis_road (Table 3). However, positive correlations between ESs and the former two urbanization indicators, as well as negative correlations between ESs changes and Dis_road were observed in GWR results. For example, positive correlations between GP and PG were mainly found in mountainous areas in the northeast and northwest (Figure 6). Positive correlations between CS and ULE were mainly distributed in plain areas in the countryside (Figure 7). Negative correlations between CS and Dis_road were observed in mountainous areas in the northwest. We claimed that some factors other than urbanization may contribute to such unexpected results, which involve natural factors (e.g., climate, soil type, terrain, land cover and hydrology), social activities (e.g., constructions of agricultural facilities and water conservancy facilities) and some methodological aspects (e.g., data preprocessing, the selection of grid size and determination of kernel bandwidth).

#### 3.3.4. Model Performance of GWR and its Comparison with OLS’s

Comparisons of model performance between GWRs and OLSs were displayed in Table 3 and Table 4. The higher adjusted R^2^ indicates higher explanatory power and the lower AICc value signifies a more concise model. We can observe dramatic improvements in adjusted R^2^ values and decreases in AICc values of GWRs over OLSs, indicating better model performance of GWR over OLS. Moreover, GWRs at 5 km grid scale obtained higher adjusted R^2^ values and lower AICc values than those at 10 km grid scale, signifying better performance of GWR at finer grid level.

Table 5 presents Moran’s I values of model residuals for GWR and OLS. Moran’s I of model residuals examines spatial autocorrelation in model residuals, which manifests the ability of model to address the spatial autocorrelation issues [40]. We can see significant positive autocorrelations of model residuals for most OLSs, on the contrary, almost no significant autocorrelations for GWRs. In addition, almost all GWR models presented dramatic decrease in Moran’s I values of their residuals compared to the corresponding OLSs. The results demonstrated that spatial autocorrelation issues could be addressed by GWR to some extent.

## 4. Discussion

### 4.1. Scale Effects in Spatially Non-Stationary Responses of Ecosystem Services to Urbanization

Scale effect often arises in spatial relationships, which refers to a spatial or geographic phenomenon usually changes with the scale of analysis (unit or level) [45]. Scale effect was closely linked to ESs’ assessment, mapping, drivers identification and interactions of with human activities [46]. This study detected two kinds of scale effects in the spatial interactions between ESs and urbanization: effect of different bandwidths and effect of different grid levels.

Effect of bandwidths mainly refers to that the stationary index of GWR declined rapidly as the bandwidth increased (see evidence in Figure 5). This may be due to, as the bandwidth increases, GWR becomes closer to global regressions, thus generating more stable spatial patterns and less spatial heterogeneity. However, there is a threshold in the curve, above which the decline of the non-stationarity becomes smooth. In this study, thresholds for ULE at 5 km and 10 km were 15,000 and 30,000 m, respectively. The threshold bandwidth can be regarded as the optimal spatial extent, within which ESs’ responses to an urbanization activity should be evaluated.

Effect of grid scales mainly refers to that GWR model performance varied at different grid scales. GWR outperformed OLS at both scales (Table 3, Table 4 and Table 5) and GWR had better model performance at finer scale (Table 3). Moreover, the concise level (Table 4), the spatial autocorrelation of model residual (Table 5), the spatial pattern of local estimates (Figure 6, Figure 7, Figure 8 and Figure 9) were all differed with scales. Generally speaking, GWR had better performance and thus was more effective to explain the spatially varying relationships at 5 km grid scale. The effect of grid scales emphasizes the analysis of socio-ecological interactions on a fine scale, which also calls for the refined evaluation of socio-ecological factors.

### 4.2. Implications for Ecologically Friendly Urban Planning

The ecological effect of urbanization should be considered during urban planning [47]. ESs, as the production of ecological processes and functions, are vulnerable to human disturbance, and changes of them can effectively reflect the human impacts on ecosystems [48]. This research mapped the dynamics of ESs, quantified the local responses of ESs to urbanization and discussed the scale effect in the ESs-urbanization interactions. It is implicative to the evaluation of urban impacts, the rational arrangement of urban activities and the formulation of ecological policies. The following section presents the specific conclusions in this study and their guidance for eco-friendly urban planning.

First of all, mapping of ESs changes helps to detect ecological degradation. Areas with significant ESs decline are required to allocate some ecological restoration projects. For example, authorities and planners could design some green infrastructure, e.g., a country park or urban boulevards, in an area with significant CS degradation. Similarly, in GP degradation areas, farmers are suggested to take some measures to improve the productivity of farmland, e.g., improving irrigation condition and updating crop varieties.

Second, global responses of different ESs to urbanization (see OLS results in Table 2) delivery information about the sensitivities of these services to urban development. The sensitivities of different ESs could then be referred as the objective importance of them to be protected during urban construction. Planners could then utilize the objective importance criteria to weight multiple ESs on a land patch to calculate the resistance of this patch to urban expansion [49,50]. To be specific, planners should highlight the most sensitive services and give them highest weights. This objective weighting criteria supplements the subjective criteria in previous studies [51,52], which determined weights according the perception of stakeholders.

Third, local responses of ESs to urbanization provide detailed information for location-based ecological assessment and site selection of construction activities. Planners and managers could obtain a set of location-based coefficients to indicate the potential ecological responses to urban construction. When they are attempting to locate an urban activity, they could compare the values of potential ecological response at different places and then chose the optimal site where less ecological degradation would be caused. Moreover, GWR model also facilitates to explore the environmental settings affecting the local relationships, like natural, economic and social factors. In areas with ESs severely threatened by urbanization, for example, in urban infringes, improving environmental background (e.g., constructing green infrastructure and reducing the building ratio) may help to relieve urban pressures and enhance human wellbeing.

Finally, scale analysis helps to obtain comprehensive effects of urbanization on ESs at multiple scales and examine the consistency of the results. Moreover, sensitivity of non-stationarity to kernel bandwidth signifies the maximum spatial extent of urban impacts. If a place locates out of the extent, the influence of urbanization on this place could be ignored.

### 4.3. Strengths and Limitations

Different from previous studies, this research took the spatial characteristics of ESs changes into consideration by using GWR to model ESs changes. GWR could produce a series of local parameter estimates, which gives detailed information for local decision making. Moreover, this study was conducted at multiple scales. Cross-scale comparison was implemented to discuss the scale effect in the relationships between ESs and urbanization. Scale analysis helps to improve the authenticity of modeling result and improves our understanding on the intrinsic processes underlying the interactions between ESs and urbanization.

However, some drawbacks still exist in this research. For example, urbanization can be a complex process concerning social, economic and cultural dimensions. Indicators selected in this study could only partly represent urbanization. With this regard, further study could develop a more comprehensive indicator system to measure urbanization, for example, incorporate energy consumption and human activities into consideration. Furthermore, spatial relationships between ESs changes and urbanization were only analyzed at two grid scales. We are not able to detect how the relationships change at various grid scales and whether there exists an optimal level for analysis. Therefore, a continuous series of spatial scales should be further adopted to obtain more reliable and comprehensive results.

## 5. Conclusions

This study utilized geographically weighted regression (GWR) to explore the spatially varying and scale-dependent responses of ESs to urbanization. Spatial patterns of changes for four ESs (i.e., GP, CS, BP and EP) showed uneven distributions across the region and featured “global similarity and local difference” across different scales. Moran’s I analysis revealed that clustering characteristics commonly existed in the spatial distributions of ESs changes. Due to the un-even distributions of ESs changes, GWR was proven to be powerful than OLS in interpreting their relationships with urbanization, since GWR produced higher R^2^, lower AICc values and spatial autocorrelations in model residuals. GWR results showed that direction and strength of impact urbanization exerting on ESs varied across space. Moreover, the spatially non-stationary relationships were scale-dependent. Two types of scale effects were identified: effects of different bandwidths and effects of different grid scales. Specifically, the stationary index of GWR declined rapidly as the bandwidth increased until reaching to a distance threshold. In addition, GWR outperformed OLS at both grid scales, and model performance of GWR was better at finer grid level.

This study is meaningful for local urban landscape planning and management. Visualization of ESs changes allows urban planners to detect ecological degradation/improvement areas. GWR model helps to obtain location-based ecological effects of urbanization, and contributes to the site selection of urban activities. Scale analysis helps to find the maximum spatial extent within which urbanization induced ecological change; it also emphasizes fine-scale analysis. In conclusion, findings in this study could provide a scientific basis for ecologically friendly urban planning and policy-making.

## Figures and Tables

**Figure 1 ijerph-17-02989-f001:**
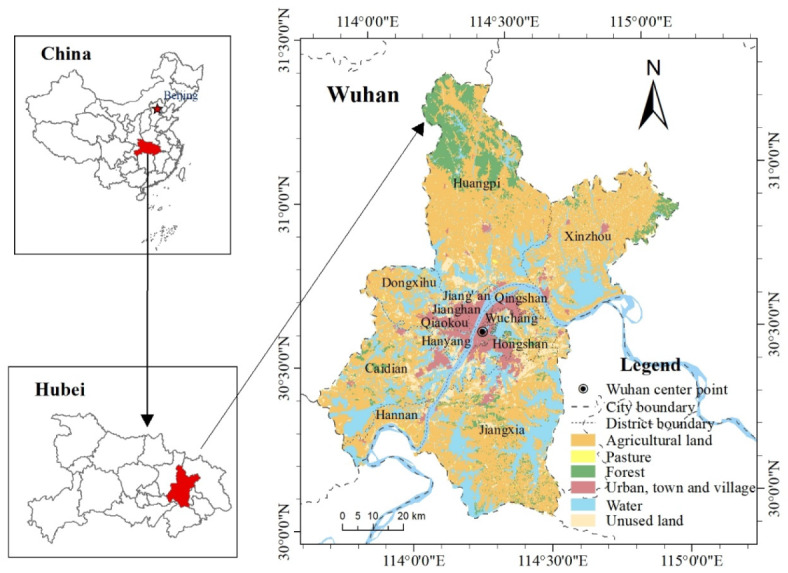
Location of the study area.

**Figure 2 ijerph-17-02989-f002:**
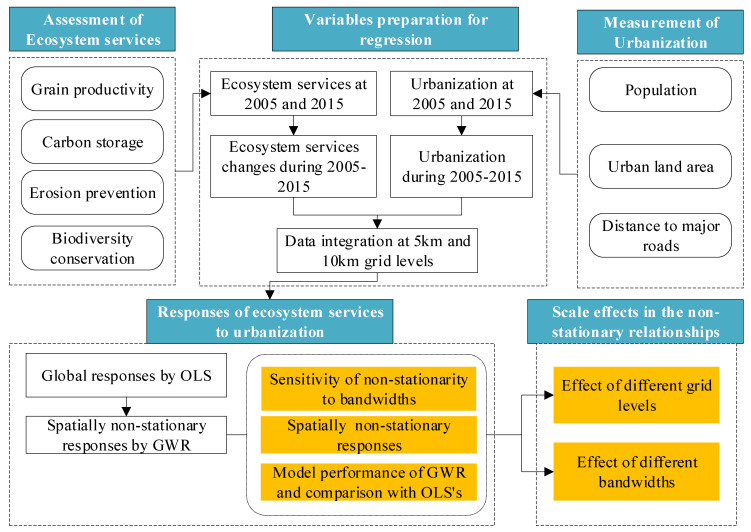
Flow chart of the method.

**Figure 3 ijerph-17-02989-f003:**
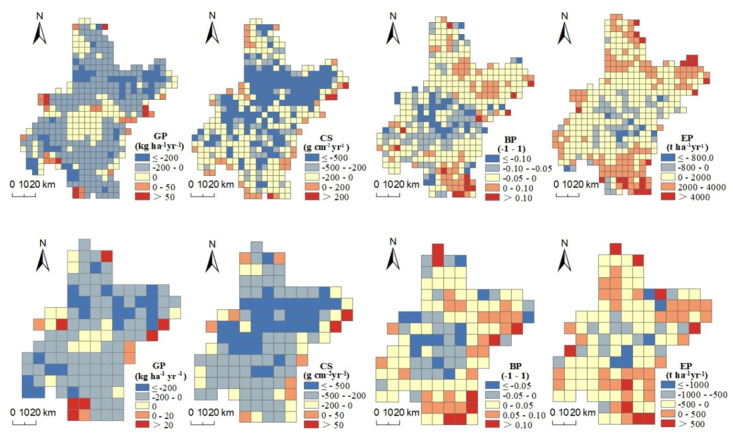
Maps of ecosystem services changes during 2005–2015 at 5 km (**upper row**) and 10 km (**lower row**) grid scales. Abbreviations: Grain productivity (GP); Carbon sequestration (CS); Biodiversity potential (BP); Erosion prevention (EP).

**Figure 4 ijerph-17-02989-f004:**
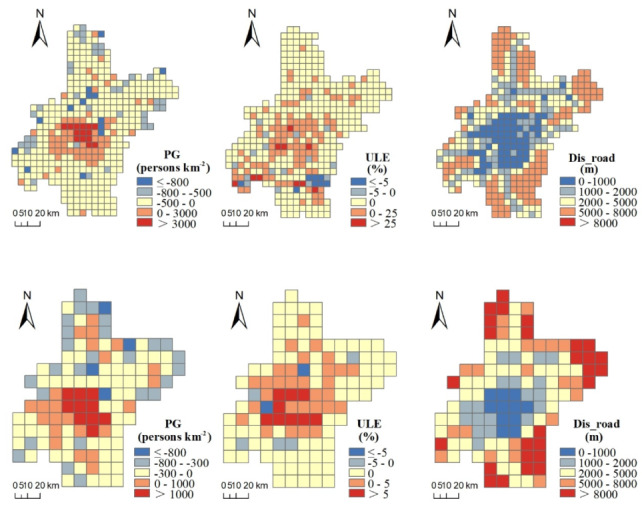
Spatial patterns of indicators for urbanization during 2005–2015 at 5 km (**upper row**) and 10 km (**lower row**) grid scales. Abbreviations: Population growth (PG); Urban land expansion (ULE); Distance to major roads (Dis_road).

**Figure 5 ijerph-17-02989-f005:**
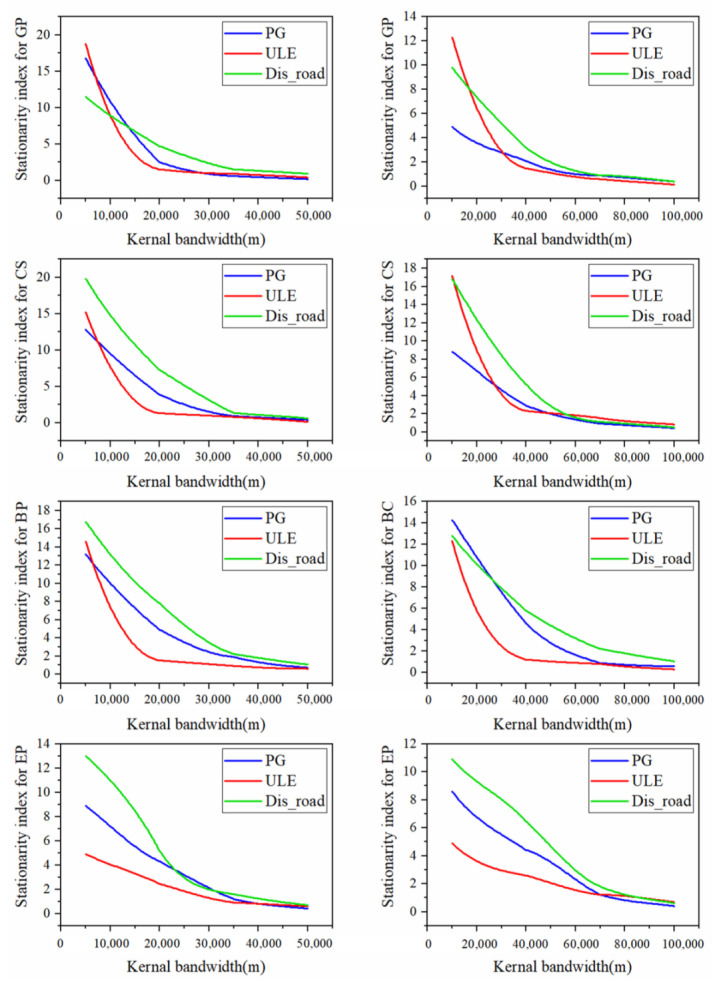
Correlogram between stationarity index and kernel bandwidth in GWR at 5 km (**left column**) and 10 km (**right column**) grid scales. Abbreviations: Grain productivity (GP); Carbon sequestration (CS); Biodiversity potential (BP); Erosion prevention (EP); Population growth (PG); Urban land expansion (ULE); Distance to major roads (Dis_road).

**Figure 6 ijerph-17-02989-f006:**
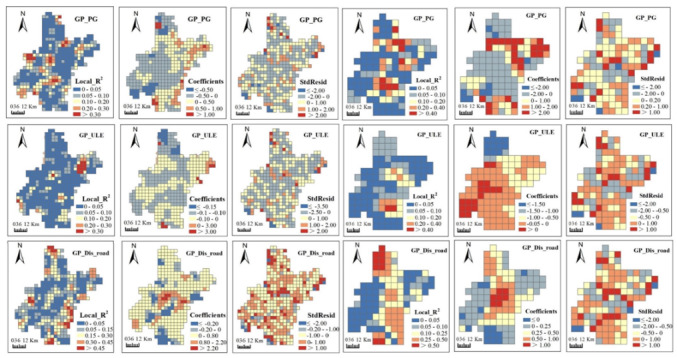
Spatial patterns of local R^2^, coefficients and standard residuals (StdResid) from geographically weighted regressions (GWRs) between grain productivity (GP) and urbanization indicators at 5 km (**left three columns**) and 10 km (**right three columns**) grid scales. Abbreviations: Population growth (PG); Urban land expansion (ULE); Distance to major roads (Dis_road).

**Figure 7 ijerph-17-02989-f007:**
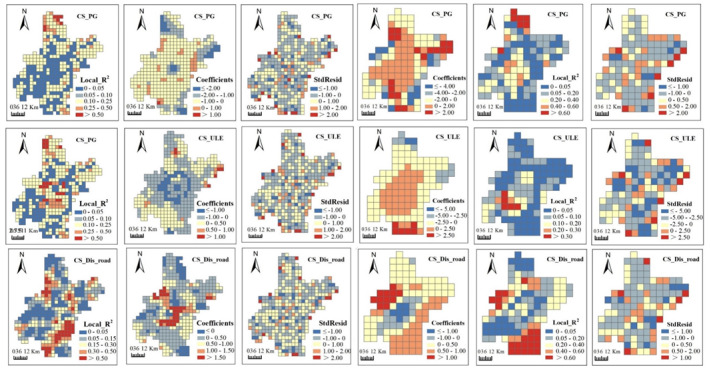
Spatial patterns of local R^2^, coefficients and standard residuals (StdResid) from geographically weighted regressions (GWRs) between carbon sequestration (CS) and urbanization indicators at 5 km (**left three columns**) and 10 km (**right three columns**) grid scales. Abbreviations: Population growth (PG); Urban land expansion (ULE); Distance to major roads (Dis_road).

**Figure 8 ijerph-17-02989-f008:**
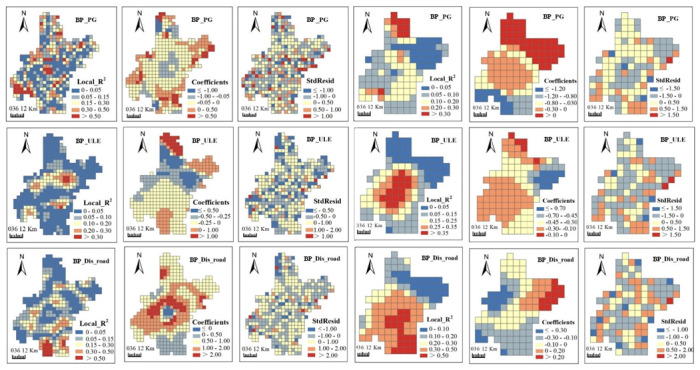
Spatial patterns of local R^2^, coefficients and standard residuals (StdResid) from geographically weighted regressions (GWRs) between Biodiversity potential (BP) and urbanization indicators at 5 km (**left three columns**) and 10 km (**right three columns**) grid scales. Abbreviations: Population growth (PG); Urban land expansion (ULE); Distance to major roads (Dis_road).

**Figure 9 ijerph-17-02989-f009:**
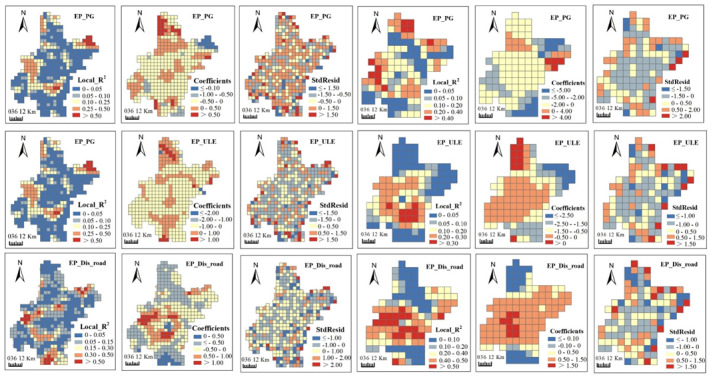
Spatial patterns of local R^2^, coefficients and standard residuals (StdResid) from geographically weighted regressions (GWRs) between erosion prevention (EP) and urbanization indicators at 5 km (**left three columns**) and 10 km (**right three columns**) grid scales. Abbreviations: Population growth (PG); Urban land expansion (ULE); Distance to major roads (Dis_road).

**Table 1 ijerph-17-02989-t001:** Globally spatial autocorrelations of ecosystem services changes at two grid scales during 2005–2015.

ESs	5 km Grid Level		10 km Grid Level	
	Moran’s I	*p*-Value	Moran’s I	*p*-Value
GP	0.4895 **	0.001	0.4788 **	0.001
CS	0.5032 **	0.001	0.3974 **	0.001
BP	0.1818 **	0.001	0.0155	0.153
EP	0.0685 **	0.001	0.0853 **	0.002

Abbreviations: Ecosystem services (ESs); Grain productivity (GP); Carbon sequestration (CS); Biodiversity potential (BP); Erosion prevention (EP). ** denotes significant at level *p* < 0.01.

**Table 2 ijerph-17-02989-t002:** Coefficients between ecosystem services (ESs) changes and urbanization indicators from ordinary least squares.

Grid Scales	ESs	PG	ULE	Dis_Road
5 km grid level	GP	−0.6597 **	−0.3937 **	0.4104 **
	CS	−0.1228 **	−0.192	0.4372 **
	BP	−0.1552 **	−0.3391 *	0.1109 **
	EP	−0.5527 **	−0.3114 **	0.2925 **
10 km grid level	GP	−0.4105 **	−0.3382 **	0.3768 **
	CS	−0.1068 *	−0.1092 *	0.3473 **
	BP	0.1307	−0.1418 **	0.0764
	EP	−0.5846 **	−0.2898 **	0.5109 **

Abbreviations: Grain productivity (GP); Carbon sequestration (CS); Biodiversity potential (BP); Erosion prevention (EP); Population growth (PG); Urban land expansion (ULE); Distance to major roads (Dis_road). ** denotes significant at level *p* < 0.01, * denotes significant at level *p* < 0.05.

**Table 3 ijerph-17-02989-t003:** Adjusted R^2^ values of ordinary least squares (OLSs) and geographically weighted regressions (GWRs) for the relationships between ecosystem services changes and urbanization indicators.

Grid Scales	ESs	PG		ULE		Dis_Road	
		Adjusted R^2^(g)	Adjusted R^2^(o)	Adjusted R^2^(g)	Adjusted R^2^(o)	Adjusted R^2^(g)	Adjusted R^2^(o)
5 km grid	GP	0.7852	0.1461	0.7667	0.1305	0.7780	0.1374
	CS	0.7667	0.1433	0.5157	0.1044	0.5527	0.2138
	BP	0.5459	0.1164	0.2703	0.1275	0.3261	0.0311
	EP	0.6039	0.1581	0.6009	0.1385	0.6347	0.1387
10 km grid	GP	0.6212	0.1479	0.6721	0.1409	0.6185	0.1284
	CS	0.5025	0.1083	0.4259	0.1012	0.4298	0.1439
	BP	0.3659	0.1075	0.1582	0.1301	0.3961	0.1229
	EP	0.5546	0.1120	0.4819	0.1686	0.6182	0.2422

Abbreviations: Ecosystem services (ESs); Grain productivity (GP); Carbon sequestration (CS); Biodiversity potential (BP); Erosion prevention (EP); Population growth (PG); Urban land expansion (ULE); Distance to major roads (Dis_road). Adjusted R^2^(g) denotes Adjusted R^2^ value for GWR model, while Adjusted R^2^(o) denotes Adjusted R^2^ value for OLS model.

**Table 4 ijerph-17-02989-t004:** The corrected Akaike information criterion (AICc) values of ordinary least squares (OLSs) and geographically weighted regressions (GWRs) for the relationships between ecosystem services changes and urbanization indicators.

Grid Scales	ESs	PG		ULE		Dis_road	
		AICc(g)	AICc(o)	AICc(g)	AICc(o)	AICc(g)	AICc(o)
5 km grid	GP	−575.3537	−42.0953	−547.3657	−35.5334	−565.1398	−81.9734
	CS	−388.1993	−60.6587	−387.1123	−160.103	−417.0996	−258.3107
	BP	−946.9034	−819.9337	−914.41	−824.5872	−936.6223	−826.0349
	EP	−618.2621	−325.9104	−607.9558	−317.5856	−641.6658	−361.2498
10 km grid	GP	−325.1587	−3.094	−368.8998	−2.2024	−387.2158	−12.7435
	CS	−125.5125	−35.6639	−157.1558	−36.521	−213.0158	−55.4822
	BP	−258.1158	−122.3117	−264.1596	−138.24	−736.1985	−124.19
	EP	−236.8857	−17.8764	−128.1158	−25.845	−1167.6618	−37.052

Abbreviations: Ecosystem services (ESs); Grain productivity (GP); Carbon sequestration (CS); Biodiversity potential (BP); Erosion prevention (EP); Population growth (PG); Urban land expansion (ULE); Distance to major roads (Dis_road). AICc (g) denotes AICc value for GWR model, while AICc (o) denotes AICc value for OLS model.

**Table 5 ijerph-17-02989-t005:** Moran’s I of model residuals for ordinary least squares (OLSs) and geographically weighted regressions (GWRs) for the relationships between ecosystem services changes and urbanization indicators.

Grid Scales	ESs	PG		ULE		Dis_Road	
		Moran’s I of Residuals(g)	Moran’s I of Residuals(o)	Moran’s I of Residuals(g)	Moran’s I of Residuals(o)	Moran’s I of Residuals(g)	Moran’s I of Residuals(o)
5 km grid	GP	0.0442	0.6587 **	0.0409	0.7427 **	0.0477	0.7237 **
	CS	0.0645	0.4218 **	0.0798	0.4169 **	0.0491	0.3638 **
	BP	−0.0990	0.2368 **	0.042	0.3074 **	−0.0132	0.3128 **
	EP	0.0065	0.5243 **	0.0244	0.5367 **	0.0376	0.5042 **
10 km grid	GP	0.0687	0.6374 **	0.0029	0.6127 **	0.0698	0.617 **
	CS	0.0512 *	0.3310 **	0.0514	0.3196	0.0584	0.3478 **
	BP	0.0125	0.1514 **	0.0841 *	0.1391	−0.0189	0.2475 **
	EP	0.0089	0.4614 **	0.0358	0.3477 **	−0.0280	0.4064 **

Abbreviations: Ecosystem services (ESs); Grain productivity (GP); Carbon sequestration (CS); Biodiversity potential (BP); Erosion prevention (EP); Population growth (PG); Urban land expansion (ULE); Distance to major roads (Dis_road). ** denotes significant at level p < 0.01, * denotes significant at level p < 0.05. Moran’s I of residuals(g) denotes Moran’s I of residuals for GWR, while Moran’s I of residuals(o) denotes Moran’s I of residuals for OLS.

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
