# Peer review of "Exploring Spatially Non-Stationary and Scale-Dependent Responses of Ecosystem Services to Urbanization in Wuhan, China"

_ijerph, 2020, doi:10.3390/ijerph17092989_

Round 1
Reviewer 1 Report
The paper presents some solid works and should be merited. However, the weakness lies in its literature review and the policy discussions (as it is more of an applied study than methodological work). The literature research of this work focuses on the urbanization impact study on ecosystem services, but from what I see, the study discusses more about the relationship between urbanization/ecosystem—whether ecosystem provides key services to urbanization at different scales and how they are affected. Thus, a large chunk of literature is ignored in terms of social-ecological system/linkage and supply-demand evaluation of ecosystem services, such as:
https://link.springer.com/article/10.1007/s13280-019-01290-y
https://www.sciencedirect.com/science/article/abs/pii/S0264837709001288
https://www.sciencedirect.com/science/article/abs/pii/S1470160X11001907
The other issue is that the practical implications of the paper is not well connected to the conclusion. After several reading, I still cannot figure out how your findings of “spatial heterogeneity” really helps urban planning and ecological conservation rather than a spatial statistic practice? The planning and policy implications, are generic at best, and can be imagined without reading the study result. Do you find any traditional planning methods or ecological zoning practice that can be improved by your method? What is their inadequacy and how do you address that? Or do you propose any new policy instrument to be added to the traditional methods? Some of the related discussion of spatial heterogeneity and improved planning discussions can be found at:
https://www.tandfonline.com/doi/abs/10.1080/00343404.2019.1701186
https://academic.oup.com/joeg/article-abstract/17/3/547/2930549
Line 9-10: This is not accurate statement.
Line 27: I believe this should be “MEA” not “MA”.
Line 82: Detecting spatial non-stationarity should not be the strength of GWR in this study. Jiangbo’s early study detected non-stationarity in landscape fragmentation index, which is an index with interest to see if it is spatial non-stationary. But for ecosystem service values, it is easy to see that it is not a spatial stationary process. This issue presents in many studies that the true applicability of GWR is not stated correctly.
Line 160: Spatial autocorrelations of ESs changes are too obvious because urbanization is usually continuous spatially. I do not see much information provided from such tests.
Line 178: The local estimation function needs to be specified or it is hard to see where W matrix is applied in the function.
Also, I understand that the land use change/urbanization data are acquired from multiple periods. Here the authors fail to explain what is the period of y measured in relation to x (a cross-period investigation would be useful if not done in this study).
Figure 2: Year stamp should be labelled on figures.
Line 306: I highly suspect that this could be simply explained by “restoration”. CS and urbanization has complex relationships beyond just “how well the ecological areas are preserved/restored” and it is highly dependent on the measure of CS. For example, plant-age is a key determinant of CS, is your method able to capture that?
Line 405: I believe your study does not highlight such results.
Line 438: How does “local stakeholders could accordingly downscale the central policy” relates the findings of your study?
Line 475: The implication of “accordingly choose location-suitable construction activities” is too far-stretched as it is hard to see how this study’s results can inform construction site selection (ecosystems service mapping does help, but I believe that is not the core idea of your work).
Finally, hope everything gets better at Wuhan!
Author Response
Response to Reviews 1:
Point 1: The paper presents some solid works and should be merited. However, the weakness lies in its literature review and the policy discussions (as it is more of an applied study than methodological work). The literature research of this work focuses on the urbanization impact study on ecosystem services, but from what I see, the study discusses more about the relationship between urbanization/ecosystem—whether ecosystem provides key services to urbanization at different scales and how they are affected. Thus, a large chunk of literature is ignored in terms of social-ecological system/linkage and supply-demand evaluation of ecosystem services, such as:
https://link.springer.com/article/10.1007/s13280-019-01290-y
https://www.sciencedirect.com/science/article/abs/pii/S0264837709001288
https://www.sciencedirect.com/science/article/abs/pii/S1470160X11001907
Response: Many thanks for your valuable comments, which help a lot in improving the paper’s quality. I totally agree with you on the drawback in the literature review. I downloaded the references you mentioned and added some others, to extend the scope of the literature not limited to “urbanization’s impact on ESs” but extended to “social-ecological system/linkage and supply-demand evaluation of ecosystem services”. I hope after the revision, the reference section would be more coherent to the current research topic. The corresponding revisions could be seen in Lines 55-62.
Points 2: The other issue is that the practical implications of the paper is not well connected to the conclusion. After several reading, I still cannot figure out how your findings of “spatial heterogeneity” really helps urban planning and ecological conservation rather than a spatial statistic practice? The planning and policy implications, are generic at best, and can be imagined without reading the study result. Do you find any traditional planning methods or ecological zoning practice that can be improved by your method? What is their inadequacy and how do you address that? Or do you propose any new policy instrument to be added to the traditional methods? Some of the related discussion of spatial heterogeneity and improved planning discussions can be found at:
https://www.tandfonline.com/doi/abs/10.1080/00343404.2019.1701186
https://academic.oup.com/joeg/article-abstract/17/3/547/2930549
Response: Many thanks for your comments and recommendations. We have carefully checked the Discussion section on the practical implications of this study and made extensive revisions on it to make it more connected to the results. We summarized four specific implications of this study for practice: (1) mapping of ESs would be beneficial to detect ecological degradation or improvement; (2) global responses of ESs to urbanization would provide delivery information about the sensitivities of these services to urbanization, which could be then referred as weighting strategies in the sum of these services to produce ecological resistance of urban expansion; (3) local responses of ESs would be helpful to location-based ecological effect assessment and site selection of construction activities; (4) scale analysis does not only examine the consistency of the results, but also provides references for urban planning at multiple scales. Please see details in Section 5.1. We hope the revised version would be clearer to delivery our thought.
Line 9-10: This is not accurate statement.
Response: Many thanks for your comment. We revised the Abstract section and we think the revised phrases would support such statement. If you still feel confused with this issue, please feel free to contact me again.
Line 27: I believe this should be “MEA” not “MA”.
Response: Many thanks for your comment. I have checked the relevant references like Díaz et al., 2018 Science 359 (6373), 270-272, Burkhard, B., et al., 2012 Ecological Indicators 21, 17-29. These referenced utilized “MA” as the abbreviation of “Millennium Ecosystem Assessment”. Thus, I think both “MEA” and “MA” are appropriate and I maintained it (Line 26). If you still feel confused with this issue, please feel free to contact me again.
Line 82: Detecting spatial non-stationarity should not be the strength of GWR in this study. Jiangbo’s early study detected non-stationarity in landscape fragmentation index, which is an index with interest to see if it is spatial non-stationary. But for ecosystem service values, it is easy to see that it is not a spatial stationary process. This issue presents in many studies that the true applicability of GWR is not stated correctly.
Response: Many thanks for your comment. Yes, spatial heterogeneity of ESs is obvious, but this study mainly focused on the spatial non-stationarity in responses of ESs to urbanization. We wanted to detect whether the responses of ESs varied at different locations and these responses differed spatially. Moreover, we focused on detecting the threshold at which the spatial non-stationarity appeared most obviously. Therefore, we think the application of GWR would be appropriate for this study.
Line 160: Spatial autocorrelations of ESs changes are too obvious because urbanization is usually continuous spatially. I do not see much information provided from such tests.
Response: Many thanks for your comment. Yes, spatial autocorrelation is obvious. Actually, I did test the spatial autocorrelation using Moran’s I method, see Section 3.2 for methodology and Table 1 for results. I also knew there were spatial statistical techniques (e.g., spatial lag/error model) suitable to address spatial autocorrelation. But we did not take such models because we mainly focused on the spatially varying relationships rather than constant/global ones. More importantly, we found when we implemented GWR, Moran’s I values of model residuals become almost no significant and very low. Therefore, I think GWR is effective to some extent to address spatial autocorrelation in this case. If you still feel skeptical with this question, please feel free to contact me again.
Line 178: The local estimation function needs to be specified or it is hard to see where W matrix is applied in the function.
Response: Many thanks for your suggestion. We have added the detailed explanation on the W matrix in the new manuscript. Please see Lines 229-234.
Also, I understand that the land use change/urbanization data are acquired from multiple periods. Here the authors fail to explain what is the period of y measured in relation to x (a cross-period investigation would be useful if not done in this study).
Response: Many thanks for your comment. As you suggested, we, indeed, conducted cross-period investigation with y measuring change of each individual ES during 2005-2015 while x as indicator of urbanization during 2005-2015. To make it clearer, we rephrased the section 3.2 in “Methodology” to clarify the calculation of ESs changes. In addition, we added a section 3.1 to manifest the flow chart of the method. We also made extensive revisions throughout the manuscript to make time period clearer. Please see Lines 184-185, 203, 209, 235, 258, 281.
Figure 2: Year stamp should be labelled on figures.
Response: Many thanks for your suggestion. Fig.2 (which is now named as Fig.3) actually shows the ESs changes during 2005-2015, so I think there may be no need to label the year 2005 or 2015 on the figure. We are sorry that we did not expressed accurately on the research time period.
Line 306: I highly suspect that this could be simply explained by “restoration”. CS and urbanization has complex relationships beyond just “how well the ecological areas are preserved/restored” and it is highly dependent on the measure of CS. For example, plant-age is a key determinant of CS, is your method able to capture that?
Response: Many thanks for your comment. We utilized Carnegie-Ames-Stanford Approach (CASA) model to estimate carbon storage, which takes vegetation type, NDVI, temperature, precipitation, etc. as model inputs. Plant-age is not directly considered in this model. To avoid ambiguity, we have removed the interpretation of results in Line 352 in the new manuscript.
Line 405: I believe your study does not highlight such results.
Response: Many thanks for your comment. We have revised the statement to “First of all, mapping ESs changes, as a form of ecological assessment and monitoring, is beneficial to detect ecological degradation/improvement and locate ecological restoration projects.” in Lines 442-443. We hope the revised phraseology would be more accurate.
Line 438: How does “local stakeholders could accordingly downscale the central policy” relates the findings of your study?
Response: Many thanks for your comment. We carefully checked the statement and found that they were unconnected to the results in the study, so we removed it and the relevant contents from the manuscript. Please see Lines 475.
Line 475: The implication of “accordingly choose location-suitable construction activities” is too far-stretched as it is hard to see how this study’s results can inform construction site selection (ecosystems service mapping does help, but I believe that is not the core idea of your work).
Response: Many thanks for your comment. We have rephrased it to “and benefits the site selection of urban activities.”. We hope the revised sentence would be more accurate and more coherent to the conclusions. See Lines 509-512.
Finally, hope everything gets better at Wuhan!
Response: Many thanks for your care on Wuhan. Obviously, Wuhan is becoming better and everything is getting back to normal.
Reviewer 2 Report
This is an interesting paper about the application of spatial modeling methods to investigate the relationships existing between ESs and urbanization processes. Before proceedings to publication, a few points should be addressed:
- ES concept is strongly anthropogenic, so it is bound to measurable benefits for humans. Biodiversity conservation and habitat quality, even though very relevant for ESs production, can't be considered ES. So these aspects could be included in the analysis, but not in the ES list.
- A brief rationale of the methodology, possibly integrated with a flow chart, explaining the consequentiality of the various steps, could be included at the beginning of the methods section.
- Maps readability should be improved by example including urban polygons to better understand model outputs related to the urban-rural gradient
- Results could be read and discussed also in view of the local livability and considering the perceived importance of ES and Urban services related to urbanization. In this regard, you may consider some relevant papers:
- Antognelli, S., & Vizzari, M. (2016). Ecosystem and urban services for landscape liveability: A model for quantification of stakeholders’ perceived importance. Land Use Policy, 50. https://doi.org/10.1016/j.landusepol.2015.09.023
- Antognelli, S., & Vizzari, M. (2017). Landscape liveability spatial assessment integrating ecosystem and urban services with their perceived importance by stakeholders. Ecological Indicators, 72, 703–725. https://doi.org/10.1016/j.ecolind.2016.08.015
Author Response
Responses to Reviews 2:
This is an interesting paper about the application of spatial modeling methods to investigate the relationships existing between ESs and urbanization processes. Before proceedings to publication, a few points should be addressed:
Response: Many thanks for your valuable comments and suggestions. Starting from them, we made extensive revisions on the manuscript to improve quality of the paper.
- ES concept is strongly anthropogenic, so it is bound to measurable benefits for humans. Biodiversity conservation and habitat quality, even though very relevant for ESs production, can't be considered ES. So these aspects could be included in the analysis, but not in the ES list.
Response: Many thanks for your comment. We have referred some literature and found that biodiversity is indeed not regarded as a service but the basis for ecosystem services. Therefore, I made revisions on the introduction on this service in Methodology. But as you suggested, we still maintained to analyze urbanization’s impact on biodiversity like many other ecosystem services.
- A brief rationale of the methodology, possibly integrated with a flow chart, explaining the consequentiality of the various steps, could be included at the beginning of the methods section.
Response: Many thanks for your suggestion. We have added an individual section at the beginning of the methodology section to present the procedure of the method. Besides, we gave a brief introduction on the flow chart. By doing this, we hope it would be easier to understand what we really do. Please see details in Section 3.1.
- Maps readability should be improved by example including urban polygons to better understand model outputs related to the urban-rural gradient
Response: Many thanks for your suggestion. We totally agree with you that urban-rural gradient analysis is beneficial to display the spatial varying relationships. However, in this case, we utilized regular grids as analysis unit, urban-rural gradient would cut up the regular grids. Moreover, we may not have enough space to expand such results in detail. If you still feel confused with this issue, please feel free to contact me again.
- Results could be read and discussed also in view of the local livability and considering the perceived importance of ES and Urban services related to urbanization. In this regard, you may consider some relevant papers:
Antognelli, S., & Vizzari, M. (2016). Ecosystem and urban services for landscape liveability: A model for quantification of stakeholders’ perceived importance. Land Use Policy, 50. https://doi.org/10.1016/j.landusepol.2015.09.023
Antognelli, S., & Vizzari, M. (2017). Landscape liveability spatial assessment integrating ecosystem and urban services with their perceived importance by stakeholders. Ecological Indicators, 72, 703–725.
- Response: Many thanks for your recommendation. We carefully read the papers and found that they were very enlightening to the discussion on “Integrating ESs to guide urban development” in our paper. These works explored the importance of multiple ESs perceived by stakeholders, which could be seen as subjective importance of ESs during aggregation. In contrast, our results tell more about the objective importance(/sensitivities) of ESs to urbanization, which could be seen as a supplement of what you focus on. To clarify the implications of our results from perspective of “Integrating ESs to restrict urban development”, we re-edited the relevant phraseology in Discussion section, please see details in Lines 455-460.
Reviewer 3 Report
Introduction
There needs to be a much clearer explanation of what ecosystem services are. At present, your presentation is rather superficial and sparse. For instance, you use the MA classification, but over the last fifteen years, there has been a lot of further research, development and debate on what constitutes ecosystem services and what defines them – It is therefore important to reference the work of IPBES and its conceptual model of ecosystem services - now natures contribution to people (see Díaz et al., 2018 Science 359 (6373), 270-272) and also CICES (see Common International Classification of Ecosystem Services (CICES) V5.1 Haines-Young and Potschin 2018) and how it defines and delineates ecosystem service types. Acknowledging that academic and policy debate, and evolution of thought around ecosystem services is important, as it provides context, and furnishes the reader with an understanding of the rationale behind your research. This doesn’t need to be extensive, in the sense of lots of additional text – but sufficient enough to provide a greater scope of understanding and awareness that ecosystem services as a concept and policy tool is dynamic and has undergone a change of framing.
Given that you are locating your study in China, and there is a large literature from China concerning development impacts on ecosystem services (e.g. infrastructure, urbanisation etc.), this needs to be reflected much more in the introduction (see papers in Ecosystem Services, Sustainability for instance). At present, the discussion of ecosystem services, techniques to map/assess ecosystem service-human relations and impacts, and urbanisation implications for ecosystem services provision, maintenance and sustainability is rather disjointed – the coherence of the narrative needs to be stronger and clearer. I would suggest therefore focusing much more on the situation in China and highlighting in more detail the large body of work on urban development and ecosystem services in that context, and from that drawing out the key issues this body of research presents. This would not only increase the clarity and coherence of your narrative, but also present a stronger rationale for the purpose of your work and why you’re locating your study in Wuhan for instance.
You mention some work on the shortcomings of some statistical approaches to demonstrating connections between development (e.g. urbanisation) and ecosystem services, and that GWR is a better and more appropriate approaches that manages to move beyond some of the problems with OLS regression approaches, but I think the reader is still left thinking “so what?”, “How important is this issue?” and “why should I care?”. It would be better if you more firmly argued why what you’ve identified is a problem (re: previous statistical approaches), and why your study is important and novel in addressing a crucial deficit.
In relation to lines 85-91:
- The aims of the paper need to clearer – as I understand the aims of the paper are twofold: 1) assessing the impact of urbanization on ecosystem services; 2) comparing two types of regression model to identify which performs better in examining spatio-temporal changes in urbanization-ecosystem service dynamics – my suggestion would be to articulate your aims in those terms.
- Please provide a justification, because it is not at all clear from the aims of the paper, as to what ES you are going to apply your GWR approach to, and why. Similarly, it is not clear, when you are talking about urbanization, what aspects of urbanization you are considering/measuring in relation to these – at this point – undefined ES. This needs to be set out upfront – urbanization is a complex process, so having a defined sense of what aspects of you are specifically concerned with is critical. [Some of the information you provide in Section 3, re: ecosystem services and urbanization ought to be incorporated into this section for clarity]
- Moreover, in relation to point (1) What “spatiotemporal” time frames are being considered? In relation to point (3) What do you mean by “multi-level analysis”?
In general terms, as you currently present the article, it is just another mapping paper of some ecosystem services in a urban city in China – which is neither new nor novel – so you really need to make the argument for originality and criticality here – you mention a link to policy – but that connection of how your results would influence policy and planning is not clear – a fuller explanation of how that might occur could present one avenue for illustrating the importance of the research you present.
Finally, there is the issue of relevancy of the article to the Journal. Given the focus of this particular Journal, it is not clear to me how what you’re proposing connects with human/public health perspectives?
Section 2
Section 2 should be incorporated into the Methodology section. For example, Section 2.2 would make more sense if it followed section 3.1
In Section 2.2, you need to make a fuller description of these data sources – many readers will not be familiar with them, and thus it is important that you describe them in more detail. For example, what land use classes are included in the Wuhan Natural Resources Bureau database? What is the resolution of the meteorological data that you collected? What does the normalised difference vegetation index data describe?
Methodology
3.1
Justification for choice of ES needs to be more robustly argued.
GP – this is essentially ‘crop production”? What measures underpin the VCI, and how robust and accurate are these measures? This is important if you are using this index to calculate GP.
BC - Biodiversity Conservation is not an ecosystem service. “Biodiversity” is often considered a so-called “supporting” services. But biodiversity conservation is a management activity and policy tool, not an ES. Also, it’s not clear that area of habitat quality on its own is the best proxy for “biodiversity” – do you not have any more specific measures of biodiversity such as alpha, beta and gamma diversity? What determines the value attributed to Habitat Quality e.g. what does habitat suitability mean, and how do you calculate the likelihood of LULC?
CS - How does your calculation of CS account for variation in vegetation types and the associated carbon sequestration and storage capacities of different vegetation types?
EP – please expand on the different “factors” included in this calculation and their metrics
3.2
PG/PD – it’s not clear that you’re measuring population growth rather in fact changes in population density, which is a related but different measure. This needs to be clarified.
Similarly, for ULE, what comprises your measure here – estimates of the increase in the built environment? How do you derived those estimates?
3.5.
Grid scales of 5km and 10km are still very large, which has the disadvantage that it misses a lot of variation in ES changes that happens to ES provision, maintenance and distribution at much finer grain resolutions e.g. 1km patches. This would provide a much more realistic picture of ES change – why did you opt not to include a finer resolution? For example, 5km and 10km grid squares might incorporate multiple habitat patches that may have quite different variations in biodiversity which you will not register at this resolution. Similarly, the built environment and ecosystem-urban dynamics can be quite heterogenous over small ranges, which again you won’t be able to detect at these larger spatial scales.
How does your model account for the very different properties of the ES and their interactions with the urban environment that you focus on in your paper – it’s not clear to me that your model accounts for the heterogeneity and differing ES dynamics, or indeed the interrelation between ES bundles….please can you clarify whether this is the case
Results
Please avoid interpretation in the results section, and especially deriving causal inferences without any data to evidence/support the causal linkage you’re attributing to account for the data you observe, specifically – please remove the following from the Results section.
Lines: 238-239; 249-254; 290-294; 304-314
These points could be raised and debated in the Discussion section, so long as you provide additional justification for your assertions regarding the interpretation of your data.
Discussion
You say a number of times in the discussion (e.g. lines 406, 421, 423-439) that the information provided by these maps should be included in policy decision-making processes – as evidence – but you don’t say how this may work in practice. There’s a sense in which this material will somehow magical inform and enlighten policy and planning processes, in advance as a form of mitigation, (rather than looking at impacts in retrospect), to ensure greater levels of environmental sustainability in an age of increasing urbanization – but what is not clear is how (as one of many pieces of information that could contribute to planning processes) your maps could/should/would be use…I think this is an areas you should explore further. For example, how would a planner use your maps in a cost-benefit exercise for instance to weigh-up the pros and cons of a specific development that required a choice between further urban expansion or continued biodiversity conservation - deciding the possible trade-offs or synergies regarding land development (e.g. expansion of the built environment; conversion of forest or agriculture to urban settlement).
Similarly, as an extension of that point, discussing how these mapping processes and GWR could be used by policymakers and planners to improve development-lead urbanization in the context of wider sustainability policy is another aspect that I think could be explored in the Discussion. Again, it would also help to strengthen the international appeal of your research and findings.
It is still not clear after reading the manuscript as to how your research links to health specifically – a key aspect of this Journal. One suggestion would to include some extra analysis, for instance, to map certain key socio-economic (e.g. income, employment, rural-urban migration) and demographic health (e.g. disease incidence, mental health) indicators across the same spatial-temporal period and scale as your ES and Urbanization indicators and to pull out some key patterns and interactions. This would provide a much more integrated and holistic social-ecological approach that could directly relate ES with urbanization and health and wellbeing. At the moment, the health and wellbeing (inclusive of socio-economic components) is missing, which in my opinion is problematic.
Author Response
Responses to Reviews3:
There needs to be a much clearer explanation of what ecosystem services are. At present, your presentation is rather superficial and sparse. For instance, you use the MA classification, but over the last fifteen years, there has been a lot of further research, development and debate on what constitutes ecosystem services and what defines them – It is therefore important to reference the work of IPBES and its conceptual model of ecosystem services - now natures contribution to people (see Díaz et al., 2018 Science 359 (6373), 270-272) and also CICES (see Common International Classification of Ecosystem Services (CICES) V5.1 Haines-Young and Potschin 2018) and how it defines and delineates ecosystem service types. Acknowledging that academic and policy debate, and evolution of thought around ecosystem services is important, as it provides context, and furnishes the reader with an understanding of the rationale behind your research. This doesn’t need to be extensive, in the sense of lots of additional text – but sufficient enough to provide a greater scope of understanding and awareness that ecosystem services as a concept and policy tool is dynamic and has undergone a change of framing.
Response: Many thanks for advising us to add information on the evolution of ESs concept and providing such important works to refer. We have downloaded the references you mentioned and found them provide a novel perspective for this research. In addition to them, we added some others to detail the evolution of ESs concept. By doing this, we hope to provide readers with wider insight into ESs science, and more details to understand the background of this research. The corresponding revisions in the manuscript could be seen in Para 1 in Introduction section.
Given that you are locating your study in China, and there is a large literature from China concerning development impacts on ecosystem services (e.g. infrastructure, urbanisation etc.), this needs to be reflected much more in the introduction (see papers in Ecosystem Services, Sustainability for instance). At present, the discussion of ecosystem services, techniques to map/assess ecosystem service-human relations and impacts, and urbanisation implications for ecosystem services provision, maintenance and sustainability is rather disjointed – the coherence of the narrative needs to be stronger and clearer. I would suggest therefore focusing much more on the situation in China and highlighting in more detail the large body of work on urban development and ecosystem services in that context, and from that drawing out the key issues this body of research presents. This would not only increase the clarity and coherence of your narrative, but also present a stronger rationale for the purpose of your work and why you’re locating your study in Wuhan for instance.
Response: Many thanks for your comment. We have updated the references in the Introduction and reorganized the whole section following the logic of “ESs concept evolution - mapping and assessment of ESs - ecological-social driver of ESs - impact of urbanization on ESs – statistical analysis in use to quantify relationships between ESs and urbanization”. After a wide scope of literature review, we proposed that the objective of the study is to “to explore the non-stationary and scale-dependent responses of ESs to urbanization”. We hope the revised version would be clearer and more coherent to what we really did. Please see details in Introduction section. Moreover, as you suggested, we supplied literature on “the development impacts on ecosystem services in China” in Lines 79-83.
You mention some work on the shortcomings of some statistical approaches to demonstrating connections between development (e.g. urbanization) and ecosystem services, and that GWR is a better and more appropriate approaches that manages to move beyond some of the problems with OLS regression approaches, but I think the reader is still left thinking “so what?”, “How important is this issue?” and “why should I care?”. It would be better if you more firmly argued why what you’ve identified is a problem (re: previous statistical approaches), and why your study is important and novel in addressing a crucial deficit.
Response: Many thanks for your comment. We are sorry that we did not clarify why our method is novel and important. We rephrased the literature review section and emphasized that “exploring the local-based relationships between ESs and urbanization tell more than global relationships in urban landscape planning and management”. Please see more details in Lines 76-78. Moreover, we pointed out that “In addition to spatial non-stationarity, the scale dependence in responses of ESs to urbanization should be not overlooked because ESs assessment and dynamics are usually scale dependent. Multiple analysis would help to better understand causal relationships between ESs and urbanization and improve accuracy and reliability of results interpretation.” Please see more details in Lines 91-96. By doing this, we hope to conclude the research shortcoming from both practical and methodological perspectives.
In relation to lines 85-91:
- The aims of the paper need to clearer – as I understand the aims of the paper are twofold: 1) assessing the impact of urbanization on ecosystem services; 2) comparing two types of regression model to identify which performs better in examining spatio-temporal changes in urbanization-ecosystem service dynamics – my suggestion would be to articulate your aims in those terms.
Response: Many thanks for your comment. We are sorry that we did not clarify the aims of this study. We think this may be because we did not articulate the research shortcomings of previous studies. Therefore, we reorganized the literature review section. Starting from the research gap, we proposed our aim: to explore the non-stationary and scale-dependent responses of ESs to urbanization. We then developed the aim of study to three components “(1) map spatiotemporal changes of ESs and identify clustering feature of these changes; (2) explore spatially non-stationary responses of ESs to urbanization using GWR; (3) conduct multi-scale analysis on the spatially varying responses to examine the scale effect.” See more details in 97-101. We hope this revision can make the focus or aim of this study clearer.
- Please provide a justification, because it is not at all clear from the aims of the paper, as to what ES you are going to apply your GWR approach to, and why. Similarly, it is not clear, when you are talking about urbanization, what aspects of urbanization you are considering/measuring in relation to these – at this point – undefined ES. This needs to be set out upfront – urbanization is a complex process, so having a defined sense of what aspects of you are specifically concerned with is critical.[Some of the information you provide in Section 3, re: ecosystem services and urbanization ought to be incorporated into this section for clarity]
Response: Many thanks for your comment. We are sorry that we did not clarify why GWR is necessary and what urbanization tells. So, we rephrased the reference review in urbanization’s impact on ESs (Lines 69-78), to provide more detailed background in talking about urbanization. Moreover, we rephrased the reference on the spatial regression technique for the measure of relationships between ESs and urbanization, to clarify the necessity of GWR (Lines 83-91).
Moreover, in relation to point (1) What “spatiotemporal” time frames are being considered? In relation to point (3) What do you mean by “multi-level analysis”?
Response: Many thanks for your comment. We are sorry that we unnecessarily used the term “spatiotemporal” here, which may mislead you. We have removed it from the text. Moreover, we modified the phrase “multi-level analysis” to “multi-scale analysis” for better understanding.
In general terms, as you currently present the article, it is just another mapping paper of some ecosystem services in a urban city in China – which is neither new nor novel – so you really need to make the argument for originality and criticality here – you mention a link to policy – but that connection of how your results would influence policy and planning is not clear – a fuller explanation of how that might occur could present one avenue for illustrating the importance of the research you present.
Response: Many thanks for your comment. We are sorry that we did not clarify the aims of this study. We think this may be because we did not clearly introduce the background of this research and organized literature review in a logic way. Actually, this is not only a mapping paper of some ESs. The distinctive novelty of this paper is that we explored the spatially non-stationary and scale dependent responses of ESs to urbanization. To illustrate the innovation and importance of this study, we revised the Introduction by two steps: (1) we reorganized the literature review and summarized the gaps of current research from methodological and practical perspectives (see more details in Lines 69-78, 83-91). (2) On the basis of step (1), we proposed the aims of this study and developed it into three components (see more details in Lines 97-102).
Finally, there is the issue of relevancy of the article to the Journal. Given the focus of this particular Journal, it is not clear to me how what you’re proposing connects with human/public health perspectives?
Response: Many thanks for your comment. Our study focused on the field of ecosystem services, an indicator to manifest how nature benefits human wellbeing. Dynamics of ESs, especially in response to development, could help to diagnose whether and to which extend natural ecosystems are sustainable. The sustainability of ESs is rather important to human survival and welfare. From this perspective, we think our study is coherent to the interest of this journal on “Environmental Sciences and Engineering” and “Environmental Health”. Besides, we utilized spatial regression techniques and multiple scale analysis to quantify geographical relationships, which falls within the scope of “Environmental Analysis and Methods” in this journal.
Section 2
Section 2 should be incorporated into the Methodology section. For example, Section 2.2 would make more sense if it followed section 3.1
Responses: Many thanks for your suggestion. Yes, I totally agree with you that if we moved Section 2.2, it would be more readable. However, when we trying to do like this, we found it would disorder the logic of current Methodology section. Thus, we maintained Section 2 where it was placed now. If you still feel skeptical with this, please feel free to contact me again.
In Section 2.2, you need to make a fuller description of these data sources – many readers will not be familiar with them, and thus it is important that you describe them in more detail. For example, what land use classes are included in the Wuhan Natural Resources Bureau database? What is the resolution of the meteorological data that you collected? What does the normalised difference vegetation index data describe?
Response: Many thanks for your comment. We are sorry to confuse you on the source and description of the data used in this study. We have added some details on the data like LULC classification (see Lines 123-125), meteorological data (see Lines 129-130). Normalised difference vegetation index (NDVI) was utilized to calculate services of grain product and carbon storage. The details could be seen in Table A1 in Appendix.
Methodology
3.1
Justification for choice of ES needs to be more robustly argued.
Response: Many thanks for your comment. We have revised the explanations on the selection of ESs in Methodology. Please see details in Section 3.2.
GP – this is essentially ‘crop production”? What measures underpin the VCI, and how robust and accurate are these measures? This is important if you are using this index to calculate GP.
Response: Many thanks for your comment. GP is essential “grain production”, which includes the product of “rice, wheat, millet, and soybean, etc.” We extracted the statistical GP values from the Statistical Yearbook in Wuhan and spatialized this value to cover the whole cultivated areas. VCI is an index to measure the vegetation coverage. We placed the detailed of GP in Appendix Table A1 because the paper had not enough space to place such contents.
BC - Biodiversity Conservation is not an ecosystem service. “Biodiversity” is often considered a so-called “supporting” services. But biodiversity conservation is a management activity and policy tool, not an ES. Also, it’s not clear that area of habitat quality on its own is the best proxy for “biodiversity” – do you not have any more specific measures of biodiversity such as alpha, beta and gamma diversity? What determines the value attributed to Habitat Quality e.g. what does habitat suitability mean, and how do you calculate the likelihood of LULC?
Response: Many thanks for your comment. We have checked the relevant references and found that biodiversity is highly linked to ESs, but as you pointed out, it is usually recognized as a kind of ESs. Therefore, we modified the relevant phrases on biodiversity. Please see details in Lines 173-176.
Moreover, we estimated biodiversity using Habitat Quality module in InVEST model (v3.3.3). The model combines the suitability of each LULC type for natural habitat and threat of construction to each LULC to calculate habitat quality. See more details in the User’ s Book of InVest model (v3.3.3). I am not sure whether you are referring to landscape diversity when saying “alpha, beta and gamma diversity”? If this is the case, it would tell another story, very different from the present one.
CS - How does your calculation of CS account for variation in vegetation types and the associated carbon sequestration and storage capacities of different vegetation types?
Response: Many thanks for your comment. We utilized Carnegie-Ames-Stanford Approach (CASA) model to calculate CS, which takes NDVI, temperature, precipitation, radiation and vegetation types as model input. We placed the simple calculation process in Appendix Table A1. If you have more interests in the whole calculation process, please refer to the paper [Zhang, Y., Liu, Y., etc.. On the spatial relationship between ecosystem services and urbanization: A case study in Wuhan, China[J]. Science of the Total Environment,2018, 637-638:780-790]. If you still are skeptical with the calculation process, please feel free to contact me again.
EP – please expand on the different “factors” included in this calculation and their metrics
Response: Many thanks for your comment. The simple explanations on the calculation process of EP was shown in Appendix Table A1.
3.2
PG/PD – it’s not clear that you’re measuring population growth rather in fact changes in population density, which is a related but different measure. This needs to be clarified.
Response: Many thanks for your comment. Yes, we utilized the changes in population density because we utilized regular grids as spatial units and we think the density of population would be more appropriate than the population. Moreover, data of population density is available.
Similarly, for ULE, what comprises your measure here – estimates of the increase in the built environment? How do you derived those estimates?
Response: Many thanks for your comment. We extracted urban land expansion by overlaying urban areas maps in 2015 and 2005, and calculated ULE as urban expanded area in each grid. Pease see Line 209 for details.
3.5.
Grid scales of 5km and 10km are still very large, which has the disadvantage that it misses a lot of variation in ES changes that happens to ES provision, maintenance and distribution at much finer grain resolutions e.g. 1km patches. This would provide a much more realistic picture of ES change – why did you opt not to include a finer resolution? For example, 5km and 10km grid squares might incorporate multiple habitat patches that may have quite different variations in biodiversity which you will not register at this resolution. Similarly, the built environment and ecosystem-urban dynamics can be quite heterogenous over small ranges, which again you won’t be able to detect at these larger spatial scales.
Response: Many thanks for your valuable comment. This is a very interesting thought. As you said, grid scales of 5km and 10km are not small enough. Actually, the method and data allowed us to make exploration at much finer grain resolutions, e.g., 1km patches. We had made attempt to do this before. However, we found 1km grid was not an appropriate scale to show clearly the local relationships if we chose the mapping scale in Figures 6-9. Maps on this scale would be rather fragmentized and difficult to read. Therefore, I gave up this scale in this paper. Even so, I still agree with you that 1km grid is very an appropriate scale to explore local relationships in a region like Wuhan, which this is the scale indeed on which we did analysis in the paper [Zhang, Y., Liu, Y., etc.. On the spatial relationship between ecosystem services and urbanization: A case study in Wuhan, China[J]. Science of the Total Environment, 2018, 637-638:780-790].
How does your model account for the very different properties of the ES and their interactions with the urban environment that you focus on in your paper – it’s not clear to me that your model accounts for the heterogeneity and differing ES dynamics, or indeed the interrelation between ES bundles….please can you clarify whether this is the case
Response: Many thanks for your valuable comment. We are sorry we did not clarify the relationships for which spatial heterogeneity is investigated. We quantified the spatial heterogeneity of ESs’ responses to urbanization. This is different from just mapping differing ES dynamics and also different from analyzing the interrelation between ES bundles (e.g., tradeoff or synthesis). We had made extensive revisions to clarify this issue throughout the manuscript. See Section 3.1 and Lines 184-185, 203, 209, 235, 258, 281.
Results
Please avoid interpretation in the results section, and especially deriving causal inferences without any data to evidence/support the causal linkage you’re attributing to account for the data you observe, specifically – please remove the following from the Results section.
Lines: 238-239; 249-254; 290-294; 304-314
These points could be raised and debated in the Discussion section, so long as you provide additional justification for your assertions regarding the interpretation of your data.
Response: Many thanks for your comment. We have revised some statements to avoid interpretation without data support. Please see Lines 288-289, 339-343, 348-352. However, we think some other statements are appropriate because they are easy to achieve with rather simple deduction from the original data, for examples, Lines 299-304.
Discussion
You say a number of times in the discussion (e.g. lines 406, 421, 423-439) that the information provided by these maps should be included in policy decision-making processes – as evidence – but you don’t say how this may work in practice. There’s a sense in which this material will somehow magical inform and enlighten policy and planning processes, in advance as a form of mitigation, (rather than looking at impacts in retrospect), to ensure greater levels of environmental sustainability in an age of increasing urbanization – but what is not clear is how (as one of many pieces of information that could contribute to planning processes) your maps could/should/would be use…I think this is an areas you should explore further. For example, how would a planner use your maps in a cost-benefit exercise for instance to weigh-up the pros and cons of a specific development that required a choice between further urban expansion or continued biodiversity conservation - deciding the possible trade-offs or synergies regarding land development (e.g. expansion of the built environment; conversion of forest or agriculture to urban settlement).
Response: Many thanks for your comment, which is very helpful to improve the Discussion section. As you suggested here, we could utilize the maps of ESs dynamics to inform conservation policies. Moreover, we treated ESs as factors to produce resistance surface to urban development. We summarized these ideas as two specific implications for practice in Discussion section. Please see details on “: (1) mapping of ESs would be beneficial to detect ecological degradation or improvement; (2) global responses of ESs to urbanization would provide delivery information about the sensitivities of these services to urbanization, which could be then referred to weighting them when producing ecological resistance surface to urban expansion” in Lines 452-460.
Similarly, as an extension of that point, discussing how these mapping processes and GWR could be used by policymakers and planners to improve development-lead urbanization in the context of wider sustainability policy is another aspect that I think could be explored in the Discussion. Again, it would also help to strengthen the international appeal of your research and findings.
Response: Many thanks for your comment. Similar to the last question, we reorganized the implications of using GWR to explore local relationships as point (3) “local responses of ESs would be helpful to location-based ecological effect assessment and site selection of construction activities”. Please see more details in Lines 461-466.
It is still not clear after reading the manuscript as to how your research links to health specifically – a key aspect of this Journal. One suggestion would to include some extra analysis, for instance, to map certain key socio-economic (e.g. income, employment, rural-urban migration) and demographic health (e.g. disease incidence, mental health) indicators across the same spatial-temporal period and scale as your ES and Urbanization indicators and to pull out some key patterns and interactions. This would provide a much more integrated and holistic social-ecological approach that could directly relate ES with urbanization and health and wellbeing. At the moment, the health and wellbeing (inclusive of socio-economic components) is missing, which in my opinion is problematic.
Response: Many thanks for your comment. You proposed a very interesting and novel thought to directly link ESs to human health, which is enlightening for our further research. Similar to a previous response, we think the present paper is coherent to the interest of this journal on “Environmental Sciences and Engineering”, “Environmental Health” and “Environmental Analysis and Methods”, because we focused on the field of ecosystem services, which is closely related to human wellbeing. Dynamics of ESs, especially in response to development, could help to diagnose whether and to which extend natural ecosystems are sustainable. Inspiration from this research is beneficial to sustainable urban development and ecological planning.
Round 2
Reviewer 1 Report
The revision has addressed reviewer concerns and should be published.
Author Response
Many thanks for your comments, which help a lot in improving the quality of the manuscript. We also appreciate your contributions in the whole review process. We have checked the English grammar throughout the manuscript. The ambiguous statements have been rephrased and the grammatical errors have been corrected.
Reviewer 3 Report
Many thanks to the authors for addressing several of the points I raised in my initial review of this manuscript, particularly in relation to the Introduction, Aims and Methodology. The second version is much improved. However, there are still some further amendments required in order for it to be accepted for publication:
The introduction still needs, especially where the impacts of urbanisation on ES are considered, to be more firmly rooted in the context of China. A short summary paragraph describing what the research concerning urbanisation and development impacts on ES in China. At present, urban impacts are discussed in a mainly generic and global scope which is not helpful for providing the necessary context for your study.
Section 2 should be included in the Methodology Section, not separate from it.
In making the connection to human and public health, it is not sufficient to say, that just because your study focuses on ES, and ES benefit human beings, therefore your study implicitly considers human wellbeing and health as an important aspect of this research. The direct connections to human and public health need to be far more explicit still. This should be done in tow ways. First, making more of a connection to ES and public health literature in the Introduction, and secondly adding more detail about the implications of your research for human (public) health. How do the data you present provide opportunities to improve public health impacts of urbanisation.
In the Discussion section you have added more about how your research could be used to inform policy decision processes - this is welcome. However, the presentation of how this information would be utilised is somewhat naive. Referring to my previous point, more detail regarding which authorities, jurisdictions, planners, managers could utilise your data for development, infrastructure and urban policy would be more informative.
Lastly, the English grammar needs some improvement. As an example, the paragraph (lines 457-465) is difficult to interpret and thus its meaning is not clear and requires rewriting. I would suggest that the English grammar is properly checked throughout the manuscript.
Author Response
Many thanks to the authors for addressing several of the points I raised in my initial review of this manuscript, particularly in relation to the Introduction, Aims and Methodology. The second version is much improved. However, there are still some further amendments required in order for it to be accepted for publication:
Response: Many thanks for your comments, which help a lot in improving the quality of the manuscript. We took them seriously and made targeted revisions on the manuscript, mainly, the Introduction and Discussion sections. Thanks again for your contributions in the whole review process.
The introduction still needs, especially where the impacts of urbanisation on ES are considered, to be more firmly rooted in the context of China. A short summary paragraph describing what the research concerning urbanisation and development impacts on ES in China. At present, urban impacts are discussed in a mainly generic and global scope which is not helpful for providing the necessary context for your study.
Response: Many thanks for your advice. We checked the manuscript and found that, as you commented, the introduction on urban impact is too generic and broad. Therefore, we amended the 3th paragraph to remove some irrelevant contents. An then we added a new 4th paragraph to detail urban development situation and urban impact studies in China. After these revisions, we found the Introduction was too long, so we removed the previous 2th paragraph. We hope the revisions would provide more details on the context of this research in Wuhan, China.
Section 2 should be included in the Methodology Section, not separate from it.
Response: Many thanks for your suggestion. We have moved the corresponding contents to the previous Methodology Section and renamed the section as “Materials and methods”.
In making the connection to human and public health, it is not sufficient to say, that just because your study focuses on ES, and ES benefit human beings, therefore your study implicitly considers human wellbeing and health as an important aspect of this research. The direct connections to human and public health need to be far more explicit still. This should be done in tow ways. First, making more of a connection to ES and public health literature in the Introduction, and secondly adding more detail about the implications of your research for human (public) health. How do the data you present provide opportunities to improve public health impacts of urbanisation.
Response: Many thanks for your suggestion. We have added some details to directly connect our study to human and public health. Please see details in Lines 21-24, 55-59 in the Introduction section and Lines 479-450 in the Discussion section.
In the Discussion section you have added more about how your research could be used to inform policy decision processes - this is welcome. However, the presentation of how this information would be utilised is somewhat naive. Referring to my previous point, more detail regarding which authorities, jurisdictions, planners, managers could utilise your data for development, infrastructure and urban policy would be more informative.
Response: Many thanks for your comment. We have amended the Discussion section to make the implications more explicit. We have removed some statements that are too generic and then developed others by giving some examples by utilizing the research results. Please see details in Discussion section.
Lastly, the English grammar needs some improvement. As an example, the paragraph (lines 457-465) is difficult to interpret and thus its meaning is not clear and requires rewriting. I would suggest that the English grammar is properly checked throughout the manuscript.
Response: Many thanks for your comment. We have checked the English grammar throughout the manuscript. The ambiguous statements have been rephrased and the grammatical errors have been corrected.